# Transcriptome Profiling of a Soybean Mutant with Salt Tolerance Induced by Gamma-ray Irradiation

**DOI:** 10.3390/plants13020254

**Published:** 2024-01-16

**Authors:** Byeong Hee Kang, Sreeparna Chowdhury, Se-Hee Kang, Seo-Young Shin, Won-Ho Lee, Hyeon-Seok Lee, Bo-Keun Ha

**Affiliations:** 1Department of Applied Plant Science, Chonnam National University, Gwangju 61186, Republic of Korea; rkdqudgml555@naver.com (B.H.K.); sreeparna@jnu.ac.kr (S.C.); wlsgml7026@naver.com (S.-H.K.); shinsy112345@naver.com (S.-Y.S.); dnjsgh201115@naver.com (W.-H.L.); 2BK21 Interdisciplinary Program in IT-Bio Convergence System, Chonnam National University, Gwangju 61186, Republic of Korea; 3National Institute of Crop Science, RDA, Wanju 55365, Republic of Korea

**Keywords:** soybean, mutant, salt tolerant, transcriptome, *GmSalt3*, phenylpropanoid pathway

## Abstract

Salt stress is a significant abiotic stress that reduces crop yield and quality globally. In this study, we utilized RNA sequencing (RNA-Seq) to identify differentially expressed genes (DEGs) in response to salt stress induced by gamma-ray irradiation in a salt-tolerant soybean mutant. The total RNA library samples were obtained from the salt-sensitive soybean cultivar Kwangan and the salt-tolerant mutant KA-1285. Samples were taken at three time points (0, 24, and 72 h) from two tissues (leaves and roots) under 200 mM NaCl. A total of 967,719,358 clean reads were generated using the Illumina NovaSeq 6000 platform, and 94.48% of these reads were mapped to 56,044 gene models of the soybean reference genome (Glycine_max_Wm82.a2.v1). The DEGs with expression values were compared at each time point within each tissue between the two soybeans. As a result, 296 DEGs were identified in the leaves, while 170 DEGs were identified in the roots. In the case of the leaves, eight DEGs were related to the phenylpropanoid biosynthesis pathway; however, in the roots, *Glyma.03G171700* within *GmSalt3*, a major QTL associated with salt tolerance in soybean plants, was differentially expressed. Overall, these differences may explain the mechanisms through which mutants exhibit enhanced tolerance to salt stress, and they may provide a basic understanding of salt tolerance in soybean plants.

## 1. Introduction

Soybean (*Glycine max* L.) is an important leguminous crop worldwide, serving as a significant source of protein and oil for human and livestock consumption [1]. It is also a multi-purpose crop utilized in the production of a wide range of industrial and consumer products, including bio-diesel, animal feed, and various processed foods [2,3]. Accordingly, soybeans are cultivated in many parts of the world, with the majority of production concentrated in the North and South Americas, including Brazil, Argentina, and the USA [4]. In particular, global soybean production for the 2023/24 season is estimated to reach 410.6 million metric tons, reflecting the increase in soybean demand.

However, soybean production is often hindered by various abiotic stressors such as drought, heat, and salinity. Among these, salt stress poses a significant threat to soybean yield [5]. Unfortunately, salt stress has been shown to have negative effects on all stages of a plant’s life cycle, including germination, vegetative growth, and the reproductive stage, resulting in the reduced yield and quality of soybeans [6,7,8]. The detrimental effects of salt stress occur when the salt concentration in the soil exceeds the permissible limit for plants, inducing morphological changes in plants through genetic, physiological, and biochemical alterations [9].

One of the major effects of salt stress is osmotic stress, and high salt concentrations in the soil reduce water availability to the roots due to osmotic imbalance [10,11]. This results in water deprivation, leading to reduced plant growth, wilting, and, ultimately, plant death. In addition, salt stress disrupts the ion transport system of the root, impairing nutrient uptake. Excessive cytoplasmic accumulation of sodium ions (Na^+^) not only disrupts normal cell function but also hinders the uptake and transport of other essential ions such as potassium, calcium, and magnesium. These disturbances induce adverse effects on various metabolic processes by impeding enzyme activity, protein synthesis, and plant growth [12,13]. In addition, high salinity in the soil inhibits the efficiency of photosystem II (PS2) and reduces the production of ATP and NADPH, which are products of phosphorylation in photosynthesis. This leads to a decrease in the photosynthetic rate and biomass accumulation [14,15].

Accordingly, plants have developed sophisticated mechanisms to survive in environments causing salt stress. Recent studies have been instrumental in identifying the salt tolerance mechanisms of plants that control ion homeostasis through the selective uptake and transport of ions [16]. However, the genetic understanding of abiotic stresses such as salt stress may be limited due to their high interaction with the environment and the complex mechanisms regulated by multiple genes [17].

Advances in high-throughput sequencing technologies, such as next-generation sequencing (NGS), are enhancing our capacity to study expression and regulation on a global scale. RNA sequencing (RNA-Seq) is widely recognized as a powerful NGS-based method enabling the comprehensive profiling of gene expression. Analysis of the transcriptome can provide new insights into the genetic and molecular bases of abiotic stress tolerance in diverse plant species [18].

Recently, several studies have contributed to the progress made in understanding the genetic and molecular mechanisms involved in the response to salt stress in soybean plants. One of these studies utilized transcriptome analysis to identify differentially expressed genes (DEGs) and their associated pathways in the roots of soybean plants. This analysis revealed that salt stress activates various stress-response genes and pathways, including those involved in ion transport, osmotic regulation, and antioxidant defense [19].

In addition, various molecular analyses of soybeans have revealed numerous genes induced by salt stress, and ongoing studies continue to explore this area [20,21]. For example, the gene *GmSOS1* (Salt Overly Sensitive 1) plays a key role in regulating ion homeostasis under salt stress by mediating Na^+^ efflux from the cytosol. The overexpression of this gene in roots has been shown to improve salt tolerance in soybean plants [22]. In addition, *GmSalt3* (Salt Tolerance 3), a gene with the potential to enhance yield under salt stress, is involved in maintaining ion homeostasis by preventing the accumulation of sodium ions (Na^+^) and chloride ions (Cl^−^) at toxic levels in plants [23,24,25].

Advances in plant breeding techniques provide advantages in the development of stress-tolerant cultivars. One of these techniques involves inducing genetic mutations through the use of physical mutagens such as gamma-ray irradiation, which can induce random mutations in the plant genome [26]. Utilization of mutants induced by gamma-ray irradiation may help in understanding the mechanisms underlying stress responses and can also be beneficial for improving productivity under adverse environmental conditions.

Previously, a salt-tolerant mutant called KA-1285 was generated from the salt-sensitive soybean cultivar Kwangan using gamma-ray irradiation [27]. The generated KA-1285 exhibited a salt-tolerance level similar to that of S-100 [28] and PI483463 [29], both of which were previously reported to be salt-resistant, based on visual evaluation under conditions of salt stress. In addition, it has been confirmed that KA-1285 possesses a major quantitative trait locus (QTL) related to salt tolerance (*GmSalt3*).

In this study, we conducted RNA-sequencing on the root and leaf tissues of two soybean genotypes (a salt-sensitive cultivar ‘Kwangan’ and its salt-tolerant mutant ‘KA-1285’) under conditions of 200 mM NaCl for 0, 24, and 72 h at the V2 stage. In addition, we conducted GO and KEGG enrichment analyses to understand the functions and pathways of differentially expressed genes induced by salt stress in the roots and leaves of salt-tolerant mutant soybean plants. Our study aims to provide new insights into the genetic and molecular mechanisms underlying salt tolerance by analyzing the transcriptome profiles of salt-tolerant soybean mutants.

## 2. Results

### 2.1. Identification of Differentially Expressed Genes Using RNA Sequencing

Thirty-six RNA library samples were obtained from the soybean cultivar ‘Kwangan’ and the salt-tolerant mutant ‘KA-1285’. Three biological replicates were taken at three time points (0, 24, and 72 h after salt treatment) from two tissues (leaves and roots) under 200 mM NaCl. These samples were analyzed using the Illumina NovaSeq 6000 platform. As a result, a total of 967,719,358 clean reads (94.48%) that passed the pre-processing stage were uniquely mapped to the soybean reference genome (Appendix A). The soybean reference genome used for analysis consisted of 56,044 genes, of which, 44,314 genes had expression values and functional annotations.

The gene expression profiles of Kwangan and KA-1285 were compared at each time point within each tissue under 200 mM NaCl (Appendix A). As a result, a total of 466 differentially expressed genes (DEGs) were identified (Table 1).

As a result, 296 and 170 DEGs were identified among the leaves and roots, respectively, and more DEGs were identified in the leaves than in the roots. Simultaneously, a comparative analysis of the DEGs was conducted based on the time point within each tissue. In the case of DEGs in the leaves, a total of 187, 60, and 49 DEGs were identified at 0, 24, and 72 h, respectively, whereas in the roots, 67, 55, and 48 DEGs were identified, respectively. Overall, the total number of DEGs decreased as the salt stress progressed in each tissue.

Next, the common DEGs identified between different time points within each tissue were compared using Venn diagrams (Figure 1; Appendix A).

Among the common DEGs identified in the leaves, 16 were upregulated, while 8 were downregulated at all time points. Similarly, in the roots, four DEGs were commonly upregulated and six were downregulated. Most of the common DEGs were identified in both the roots and leaves, but some genes showed tissue-dependent expression. Among the common DEGs identified only in the leaves, *Glyma.04G180400*, which encodes a BURP-domain-containing protein, was upregulated, whereas *Glyma.03G135600*, which encodes aconitase 2, was downregulated. However, no common DEGs were identified only in the roots.

In addition, the common DEGs at 24 and 72 h under 200 mM NaCl were analyzed to identify stress-affected DEGs. Among the common DEGs in the leaves at 24 and 72 h under salt stress, two genes were upregulated and two genes were downregulated. Representatively, *Glyma.13G073900*, which encodes protein kinase 2A, was commonly upregulated in the leaves at both 24 and 72 h. In contrast, *Glyma.03G030300*, which encodes cytochrome P450 family 83, subfamily B, polypeptide 1 (CYP83B1), was commonly downregulated. However, in the roots at the same time points, only one gene, *Glyma.17G209900*, which encodes 12-oxophytodienoate reductase 1, was downregulated.

Additionally, we analyzed the expression patterns of genes related to plant hormones and ROS to determine which genetic mechanisms were involved and affected signal transduction in response to salt stress (Appendix A). As a result, AUX- and BR-related genes were the most differentially expressed, with 1083 and 1068 genes distributed in each hormone, respectively (Appendix A). On the other hand, the smallest number of DEGs were distributed in the GA- and JA-related gene families, which comprised 175 and 280 genes for each hormone, respectively. Also, the identified ROS-related DEGs were classified into five categories according to their functional categories (Appendix A). The groups with the largest number of distributed genes were ROS production and ROS breakdown, containing 873 and 596 genes, respectively.

### 2.2. Functional Annotation of Identified DEGs Induced by Salt Stress

Gene Ontology (GO) and Kyoto Encyclopedia of Genes and Genome (KEGG) analyses were conducted using comparative groups of DEGs identified over time under salt stress within each tissue. These were carried out in order to facilitate functional understanding.

First, the DEGs identified at each time point within each tissue were assigned to GO terms according to sequence homology, classified into three main categories: biological process (BP), cellular component (CC), and molecular function (MF) (Figure 2; Appendix A).

According to the GO analysis results, the DEGs identified within each tissue under the 200-mM-NaCl condition were redundantly enriched with 195 GO terms in the leaves and 63 GO terms in the roots. In the case of the upregulated DEGs identified in the leaves, the main GO clusters were identified as representing ‘catalytic activity’ of MF (0 h), ‘phosphorylation’ of BP (24 h), and ‘transferase activity’ of MF (72 h). On the other hand, for the downregulated DEGs, ‘binding’ of MF (0 h), ‘ion binding’ of MF (24 h), and ‘response to stress’ of BP (72 h) were identified as the major clusters. Likewise, in the roots, ‘binding’ of MF (0 h) and ‘oxidoreductase activity, acting on paired donors, with incorporation or reduction in molecular oxygen’ of MF (72 h) were identified as the major clusters of downregulated DEGs. For upregulated DEGs at 24 h under salt stress, no significant GO terms were identified. On the other hand, in the case of the downregulated DEGs, the main clusters were identified as the ‘cytoplasm’ of CC (0 h), ‘catalytic activity’ of MF (24 h), and ‘ADP binding’ of MF (72 h).

In the KEGG pathway analysis (Figure 3; Appendix A), the majority of the DEGs identified, regardless of the tissue and time points, were commonly mapped to ‘metabolic pathways’ or ‘biosynthesis of secondary metabolites’. However, some of the DEGs identified in the leaves at 0 h under 200 mM NaCl were uniquely mapped to ‘phenylpropanoid biosynthesis’.

A total of nine genes related to the phenylpropanoid biosynthesis pathway were classified into four clusters according to their protein function (Table 2).

Most of the genes were classified as ‘peroxidase (EC 1.11.1.7)’, while some genes were clustered as ‘coniferyl-aldehyde dehydrogenase (EC 1.2.1.68)’, ‘caffeate O-methyltransferase (EC 2.1.1.68)’, and ‘shikimate O-hydroxycinnamoyltransferase (EC 2.3.1.133)’. However, the majority of genes related to the phenylpropanoid biosynthesis pathway were upregulated at 0 h under 200 mM NaCl in KA-1285 compared to Kwangan; however, *Glyma.20G045800* (EC 23.1.133) was downregulated overall. The role of each gene in the phenylpropanoid biosynthesis pathway is described in Figure 4.

### 2.3. The Expression Patterns of DEGs Identified in the Salt-Tolerant Mutant

Hierarchical clustering analysis was conducted to investigate expression patterns using all DEGs identified between Kwangan and KA-1285, regardless of the time points within each tissue, under 200 mM NaCl (Figure 5; Appendix A).

Clustering was conducted based on the Log_2_ (fold change, FC) of the expression levels of DEGs identified in each tissue. As a result, 228 DEGs identified in the leaves were classified into four clusters, while 125 DEGs identified in the roots were classified into six clusters. Regarding the clusters classified based on the DEGs identified in the leaves, Cluster 1 (83 genes) and Cluster 3 (43 genes) consisted of upregulated and downregulated DEGs, respectively. The clusters 1 and 3 contained, respectively, *Glyma.04G180400* encoding ‘BURP-domain-containing protein’ and *Glyma.13G056600* encoding ‘UDP-Glycosyltransferase superfamily protein’. Cluster 2 (98 genes) and Cluster 4 (4 genes) were composed of DEGs that were upregulated and downregulated, respectively, at 0 h under salt stress and commonly showed no difference between Kwangan and KA-1285 as time progressed (24 h and 72 h). Cluster 2 contained genes previously associated with the phenylpropanoid pathway (except *Glyma.20G045800*), and Cluster 4 contained *Glyma.16G110100*, which encodes ATP-binding cassette 14.

On the other hand, DEGs identified in roots exhibited a more diverse classification of expression patterns compared to those identified in leaves, but most DEGs exhibited expression patterns similar to those observed in leaves. DEGs belonging to Cluster 1 were generally upregulated in KA-1285 compared to Kwangan, but as time progressed, their expression levels decreased and increased again at 72 h under salt stress. Cluster 3 and Cluster 5 exhibited similar expression patterns, but the DEGs in Cluster 3 remained downregulated at 72 h. Clusters 1 to 6 identified in the roots contained the genes *Glyma.03G171700*, *Glyma.13G098600*, *Glyma.03G048600*, *Glyma.09G158000*, *Glyma.10G111200*, and *Glyma.19G007700*, respectively, which displayed the most dramatic expression patterns within their respective clusters. Specifically, these genes encode ‘cation/H^+^ exchanger 20’, ‘villin 4’, ‘disease resistance protein (TIR-NBS-LRR class) family’, ‘translation initiation factor IF6’, ‘trichome birefringence-like 36’, and ‘carbonic anhydrase 1’.

### 2.4. Validation of Selected DEGs

The selection of DEGs for validation was conducted through a comparative analysis between the results of RNA-Seq analysis and the genetic mutations identified between the Kwangan and the KA-1285.

A total of 10 genes from clusters in each tissue were selected to validate the results obtained through RNA-Seq analysis, and their relative expression levels were analyzed using reverse transcription followed by quantitative polymerase chain reaction (RT-qPCR) (Figure 6).

The selected DEGs from the four clusters identified in the leaves were *Glyma.04G180400* (Cluster 1), *Glyma.04G103400* (Cluster 2), *Glyma.13G056600* (Cluster 3), and *Glyma.04G168500* (Cluster 4). The selected DEGs from the six clusters identified in the roots were *Glyma.03G171700* (Cluster 1), *Glyma.17G209900* (Cluster 2), *Glyma.17G111100* (Cluster 3), *Glyma.13G035900* (Cluster 4), *Glyma.17G235000* (Cluster 5), and *Glyma.09G210900* (Cluster 6). The expression patterns of all selected genes showed consistent results compared to the RNA-Seq data.

Additionally, the RT-qPCR was conducted to validate the changes observed in phenylpropanoid biosynthesis pathway in the leaves using the 9 DEGs (Figure 7a). Most of the selected genes exhibited expression patterns consistent with the results of the RNA-Seq analysis, although there were differences in expression levels. However, *Glyma.05G231800* (EC: 1.2.1.68) showed a different expression pattern in the qPCR analysis compared to the RNA-Seq results.

Next, we compared the relative lignin contents of Kwangan and KA-1285 using the acetyl bromide method to analyze the relationship between the genes related to lignin biosynthesis in the phenylpropanoid pathway and the lignin content of Kwangan and KA-1285 (Figure 7b). The acetyl bromide soluble lignin (ASBL) in the Kwangan increased significantly as time progressed under 200 mM NaCl, but no difference was detected in the KA-1285. However, overall, the ASBL of the KA-1285 slightly increased compared to the Kwangan.

## 3. Discussion

Global climate change affects various forms of abiotic stresses in agricultural and natural ecosystems, and extensive research has been conducted to assess their effects [30]. In particular, the majority agronomic research on climate change has focused on drought conditions and salt stress, which are closely related [31]. Prolonged drought increases soil salinity, disrupts plant nutrient uptake and water balance, and ultimately reduces plant productivity [32]. Plants have developed sophisticated mechanisms to survive these adverse conditions, including differential expression of numerous genes [33]. The analysis of these transcriptomic changes has been utilized as a powerful for investigating the molecular regulatory mechanisms induced by abiotic and biotic stresses [18].

In this study, we conducted a transcriptome analysis to identify DEGs in the roots and leaves of the soybean cultivar ‘Kwangan’ and the salt-tolerant mutant ‘KA-1285’, which was induced by gamma-ray irradiation, in response to salt stress. The major salt tolerance QTL of the KA-1285 was mapped to ‘*GmSalt3*’ region on chromosome 3 [27]. This mutant exhibited reduced accumulation of Cl^−^ in leaves and increased accumulation of Cl^−^ in roots compared to the original cultivar. Through RNA sequencing, we successfully identified 466 DEGs under 200 mM NaCl, regardless of tissue and time point (Table 1). The DEGs identified between the salt-tolerant mutant and the salt-sensitive soybean were distributed throughout the entire genome. This indicates that gamma-ray irradiation may induce significant genetic changes that alter gene expression across all soybean chromosomes. In addition, it was confirmed that each DEG in the roots and leaves showed an identical expression pattern under salt stress regardless of tissue. This suggests that, at least in our study, most genes responding to salt stress are simultaneously regulated throughout the entire plant.

These results were consistent with the KEGG enrichment analysis, and most DEGs were enriched in major pathways, including ‘Metabolic pathways’ and ‘Biosynthesis of secondary metabolites, regardless of tissue (Figure 3). This suggests that salt stress induces changes in various metabolic pathways involved in salt tolerance by regulating the biosynthesis of various secondary metabolites. In fact, it is well known that environmental changes, such as diseases and abiotic stresses, affect the production of secondary metabolites of in plants, unlike primary metabolites that are related to the biological activity and growth of plants [34,35]. In summary, the salt-tolerant mutant KA-1285 activates or alters several metabolic pathways in cells and tissues at the whole plant level to produce various secondary metabolites, and thus it can be assumed that salt tolerance is enhanced through these changes. Nevertheless, a few identified DEGs were differentially expressed in an independent manner regardless of tissue, and we focused on these genes.

Representatively, eight genes related to the phenylpropanoid pathway (one gene whose expression pattern was not validated was excluded) were only differentially expressed in the leaves of the KA-1285 regardless of salt stress (0 h) (Table 2; Figure 4). These genes are mainly related to ROS production and encode members of the peroxidase family. Various stresses such as gamma rays and high soil salinity induce ROS accumulation in plants, resulting in oxidative stress. Based on this, plants induce various defense mechanisms against stress by regulating various biological processes through hormones. In this study, five ROS gene groups, including ROS production and breakdown, and eight plant hormone groups, including AUX- and BR-related genes, were regulated (Appendix A). However, our results showed a relatively greater number of growth-regulating hormones (AUX, GA, CK, BR) compared to stress-responsive hormones (ABA, ETH, SA, JA). This suggests that growth-regulating hormones regulate plant growth and development under normal as well as abnormal conditions and could induce plant adaptation and establishment of defense systems against environmental stress through crosstalk with other hormones.

Specifically, the phenylpropanoid pathway is activated by various external factors, including abiotic stress, and is known to be involved in the life cycle and stress tolerance of plants [36,37]. One of the final products of this pathway is an alcohol called monolignol, which is one of the precursors of lignin, a major component of cell walls. Lignin content is essential for plant survival, as it generally increases in roots and leaves when exposed to various stresses [38]. Based on the genetic and physiological backgrounds mentioned above, we compared the accumulation of lignin over time in the leaves of Kwangan and KA-1285 (Figure 7). As a result, we confirmed a statistically significant increase in lignin accumulation in KA-1285. We concluded that genes related to the phenylpropanoid pathway were upregulated in the KA-1285 developed from gamma-ray irradiation. Accordingly, the increased accumulation of lignin in the KA-1285 could enhance salt tolerance. Several studies have reported that various forms of radiation increase phenolic compounds and lignin-related substances in plants, which supports our results [38,39]. Indeed, lignin is involved in vascular transport as well as structural support in plants [38]. Therefore, one of the factors contributing to the improved salt tolerance in the KA-1285 might be the modulation of water movement and ions by lignin, ultimately preventing the accumulation of Cl^−^ in leaves. However, further study is still required on the crosstalk between these mechanisms elaborated yet irregularly occurring hormones.

Next, the expression patterns of the identified DEGs were investigated at three different time points (0, 24, and 72 h after salt treatment; Figure 5). DEGs identified in the leaves and roots were classified into clusters with four and six expression patterns, respectively. *Glyma.04G180400*, a gene in Cluster 1 in the leaves, belongs to the BURP-domain-containing protein family. A total of 23 BURP family genes have been found in soybean plants in a previous study, with least 14 of them reported to respond to salt stress [40]. Among these 14 genes, *Glyma.04G180400*, reported as *Gm04.3*, responds strongly to drought and ABA treatment. Other studies have confirmed its upregulation in salt-tolerant soybean lines under similar salt-stress conditions [41]. Our results confirmed its strong upregulation only in the leaves of the salt-tolerant mutant, regardless of the duration of salt treatment. This suggests that the upregulation of *Glyma.04G180400*, which encodes a BURP family gene responsive to various stress conditions, might be involved in tissue-specific stress tolerance.

*Glyma.13G056600*, a gene corresponding to Cluster 3 in the leaves, encodes a UDP-Glycosyltransferase (UGT) superfamily protein. UGT family genes are widely involved in plant growth and stress responses, and are found in a variety of plants [42,43,44]. In fact, the overexpression of various UGT genes in *Arabidopsis* has been shown to increase several stress tolerance, but some have also been reported to increase sensitivity [42]. Also, this study has reported that knockout of UGT family genes induces the upregulation of genes associated with cell-wall-related transcription factors, ultimately increasing the biosynthesis of monolignol, a precursor of lignin. Therefore, we conclude that the downregulation of *Glyma.13G056600* that we observed in the leaves induces the upregulation of genes associated with the phenylpropanoid pathway.

On the other hand, *Glyma.03G171700*, which is part of Cluster 1 in the roots, encodes the cation/H^+^ antiporter and was found to be differentially expressed only in the roots. Its expression was strongly upregulated at 0 h under 200 mM NaCl, it decreased at 24 h, and then increased again. This gene is linked to a major quantitative trait locus associated with salt tolerance, named *GmSalt3*, which was identified in a previous study. *GmSalt3* is involved in maintaining ion homeostasis under salt stress by preventing excessive ROS accumulation in the roots of plants [45]. It has also been reported to prevent Cl^−^ accumulation in leaves [24], which is consistent with our results. However, we did not identify any mutations in the exon region or within 20 kbp upstream, which is estimated to be the promoter region of *Glyma.03G171700*. Accordingly, we suggest that the expression of this gene is not directly regulated by mutations, but rather by the presence of new transcription factors (TFs). TFs are key proteins that regulate complex mechanisms in plants in response to various stresses. They help plants adapt and survive adverse environmental conditions by regulating gene expression [46]. The TFs identified in our study were distributed in 58 different families, including those known to be modulated by salt stress such as C2H2 [47], ERF [48], MYB [49], WRKY [50], and bHLH [51] (Appendix A). Based on the conventional classification of TFs into activators and repressors, we selected 25 differentially expressed genes according to their expression patterns. Among them, the TF family with the largest number of DEGs was MYB, followed by bHLH, C2H2, MYB-related TFs, and WRKY (Appendix A). This order was consistent with the overall number of TFs identified. Therefore, we conclude that some of these 25 DEGs could act as TFs that regulate *Glyma.03G171700* within the *GmSalt3* QTL associated with salt stress. However, further research is needed to understand the relationship between *GmSalt3* in the roots and the tissue-specific salt tolerance mechanisms in the leaves, which were newly identified in this study.

## 4. Materials and Methods

### 4.1. Plant Materials and Salt Treatment

In this study, we used two soybeans, ‘Kwangan’ and ‘KA-1285’, which have different tolerance levels for salt stress, for transcriptome profiling. According to the evaluation of salt tolerance in a previous study [27], KA-1285 (9.2 mg/g) showed a decrease in Cl^−^ accumulation in its leaves when compared to Kwangan (18.2 mg/g), following a two-week exposure to 200 mM NaCl. However, peculiarly, the accumulation of Cl^−^ in the roots of KA-1285 (12.8 mg/g) exhibited a slight increase compared to Kwangan (10.3 mg/g).

The sample preparation for the experiment was conducted under the following conditions. Briefly, the seeds of each soybean were sterilized with 70% ethanol for 1 min, washed, and germinated in a growth chamber at room temperature (25 °C; 16 h light/8 h dark). The germinated seedlings at the V1 stage were acclimated to a half concentration of Hoagland solution for hydroponics using 50-hole plastic trays (27 × 53 × 11.2 [h] cm); the solution was replaced every three days. Then, thirty seedlings per genotype with similar growth rates at the V2 stage were subjected to 1/2 Hoagland solution with a NaCl concentration of 200 mM. Samples for RNA isolation were obtained by randomly selecting five seedlings. The samples for each soybean were obtained at three different time points (0, 24, and 72 h after salt treatment) from two tissue types (leaves and roots), with three biological replicates. In total, thirty-six samples were rapidly frozen using liquid nitrogen and stored at –80 °C for subsequent analysis.

### 4.2. RNA Isolation, Short-Read Sequencing, and Mapping of Clean Reads

Total RNA was isolated from each sample using the RNeasy Plant Mini Kit (Qiagen, Hilden, Germany). The purity and concentration of the isolated RNA were determined using a NanoDrop™ 2000 spectrophotometer (Thermo Fisher Scientific, Waltham, MA, USA), while the quality and integrity were assessed using an Agilent 2100 Expert Bioanalyzer (Agilent, Santa Clara, CA, USA). The high-quality RNA samples were used for RNA-Seq library construction using a TruSeq RNA Library Prep Kit (Illumina, San Diego, CA, USA). Subsequently, the samples were sequenced on a NovaSeq 6000 platform (Illumina) according to the manufacturer’s guidelines (Insilicogen Inc., Youngin, Republic of Korea). The RNA-Seq dataset was deposited in the National Center for Biotechnology Information Sequence Read Archive (NCBI SRA) database: PRJNA999740, PRJNA999471, PRJNA999468, and PRJNA999461, corresponding to the roots and leaves of Kwangan and the roots and leaves of KA-1285, respectively.

Trimmomatic (v.0.39) [52] was used to remove adapter sequences and low-quality reads (phred score < 30; length of reads < 36 bp) from the raw data of the generated short reads. The cleaned short reads that passed the preprocessing stage were then aligned to the soybean reference genome (Glycine_max_Wm82.a2.v1). This was obtained from the Phytozome database (http://phytozome.jgi.doe.gov, accessed on 1 February 2023) using the HISAT2 software (v2.1.0) [53]. The gene expression values were determined based on the total number of reads mapped to each gene using the HTSeq package (v.0.11.0) [54]. Normalization was conducted using the DESeq library [55] of the R software (v4.2.1) to prevent data deviations between samples (SEEDERS Inc., Daejeon, Republic of Korea).

### 4.3. Identification and Functional Annotation Analysis of Differentially Expressed Genes (DEGs)

Among the identified genes between the treatments of each genotype, two satisfying conditions were used. Specifically, those with an adjusted *p*-value (FDR) ≤ 0.01 and a 2-fold change (|Log_2_ [fold change, FC]| ≥ 1) confirming at least a 2-fold expression change were defined as DEGs. The FDR was estimated through the procedure of Benjamini and Hochberg [56].

The information on the identified DEGs was compared with the information on genes related to ROS, hormones, and PlantTFDB v5.0 [57] using the BLAST tool (BLASTP) [58]. ROS and hormone-related genes were defined as DEGs satisfying the condition of e-value ≤ 1 × 10^−10^ and identity ≥ 40%, and Plant TFDB v5.0 was used to define genes meeting the condition of an e-value ≤ 1 × 10^−100^. Plant hormones were classified as jasmonic acid (JA), salicylic acid (SA), ethylene (ETH), auxin (AUX), cytokinin (CK), brassinosteroids (BR), abcisic acid (ABA), and gibberellin (GA). The information regarding TFs was obtained from the Plant Transcription Factor Database, which was used to identify TFs associated with the identified DEGs.

Gene ontology (GO) analysis was conducted by comparing the identified DEGs with the homology of sequences provided in the GO database [59]. The Kyoto Encyclopedia of Genes and Genomes (KEGG) analysis was conducted under the condition of e-value ≤ 1 × 10^−100^, with the best hits identified using the BLASTP for the identified DEGs and amino acid sequences from the KEGG database [60].

The information on the identified DEGs was used for the hierarchical clustering analysis using the amap [61] and gplots [62] functions of the R library. The expression patterns of DEGs were calculated using Pearson’s correlation and grouped using the complete methodology.

### 4.4. Reverse Transcription followed by Quantitative Polymerase Chain Reaction (RT-qPCR)

To validate the DEGs, we selected some DEGs exhibiting genomic variations in the coding regions of KA-1285 compared with Kwangan, which were detected through a previous re-sequencing analysis [27].

The synthesis of cDNA was conducted using the SuperScript™ III First-Strand Synthesis SuperMix kit (Invitrogen, Waltham, MA, USA) according to the procedural guidelines provided by the manufacturer.

The qPCR analysis of the synthesized cDNA was conducted using the Bio-Rad iQ™ SYBR Green Supermix kit (Invitrogen) on a StepOne Real-Time PCR System (Applied Biosystems, Foster City, CA, USA). The reaction mixture consisted of cDNA (total 20 ng) and gene-specific primers (at a concentration of 100 pmol/μL). The qPCR reaction conditions were as follows: initial incubation at 95 °C for 3 min (holding stage), followed by forty cycles of incubations at 95 °C for 15 s and at 60 °C for 60 s. A melting curve analysis was conducted to confirm the absence of PCR products and primer dimer formation. The gene-specific primers were designed using the primer-BLAST tool in NCBI [63]. The primer sets are described in Appendix A.

The relative expression levels of the selected genes were calculated using the 2^−ΔΔCt^ method [64], and each Ct value was normalized using the housekeeping gene *GmUKN1* [65]. The measured expression value was calculated as the average of two replicates, with each sample having three biological replicates.

### 4.5. Quantification of Acetyl Bromide Soluble Lignin (ABSL)

The quantification of total lignin in soybean leaves was conducted using the acetyl bromide soluble lignin (ASBL) assay [66]. For the assay, freeze-dried soybean leaves were used, and pigments such as chlorophyll were removed by treating with absolute ethanol. A 10 mg portion of the dried samples was dissolved in 2 mL of 25% acetyl bromide diluted with glacial acetic acid and immediately incubated at 70 °C for 30 min. Then, the dissolved samples were centrifuged for 1 min, and 0.9 mL of 2 M NaOH, 3 mL of glacial acetic acid, and 0.1 mL of 7.5 M hydroxylamine HCl were added sequentially. The mixtures of samples were centrifuged at 4000× *g* for 10 min, and the supernatant was diluted 20 fold with glacial acetic acid. The diluted samples were quantified using a spectrophotometer by measuring the absorbance at 280 nm, according to the method described by Barnes and Anderson. The extinction coefficient used in the formula was that of *Arabidopsis thaliana*.

### 4.6. Statistical Analysis

Statistical analysis of this study was conducted using IBM SPSS Statistics version 27 (IBM, Armonk, NY, USA). Analysis of variance (ANOVA) and the least significant difference (LSD) test (*p* < 0.05) were used to determine statistically significant differences between samples.

## 5. Conclusions

In this study, we conducted RNA sequencing to profile DEGs in response to salt stress in the transcriptomes of leaves and roots of a gamma-ray-induced salt-tolerant soybean mutant. As a result, we identified a total of 466 DEGs in the leaves and roots, and we found that most of the DEGs responded similarly regardless of the tissue type. This was also reflected in the GO and KEGG enrichment analyses, suggesting that genes involved in ‘metabolic pathways’ and the ‘biosynthesis of secondary metabolites’ may play a key role in salt stress. Nevertheless, we identified tissue-specific DEGs mapped to the phenylpropanoid pathway induced by gamma-ray irradiation in leaves. Then, we verified this result by comparing the expression levels of these genes with the accumulation of lignin, which is the final product of the pathway in the above-ground parts of plants. As a result, eight genes mapped to the phenylpropanoid biosynthesis pathway, with the exception of one, had the same expression values as with RT-PCR. Furthermore, the lignin content increased in the soybean mutant, in which these genes were upregulated. This result means that the lignin contained in the above-ground part of the plant was involved in the salt tolerance of the mutant soybean.

It is presumed that the changes in the expression patterns of these genes were caused by the downregulation of *Glyma.13G056600*, which belongs to the UGT family. This served as an opportunity to explore the tissue-specific salt-tolerance mechanism of leaves. Likewise, the root-specific expression gene *Glyma.03G171700* was identified as encoding the cation/H^+^ antiporter. This gene exists within a major QTL associated with salt tolerance, which is known to be related to Cl^−^ accumulation in above-ground plant parts, including the leaves. A noteworthy observation is that DEGs are expressed differently depending on the tissue type and time; however, this may be commonly related to ion homeostasis in above-ground plant parts, including the leaves. However, further study is needed on the interactions between different tissue-specific DEGs that exhibit similar functions in the above-ground parts of soybean plants. The results reported in this study can enhance our understanding of the overall molecular control mechanisms of salt tolerance, and they have the potential to be utilized for genetic improvement in soybean-breeding programs.

## Figures and Tables

**Figure 1 plants-13-00254-f001:**
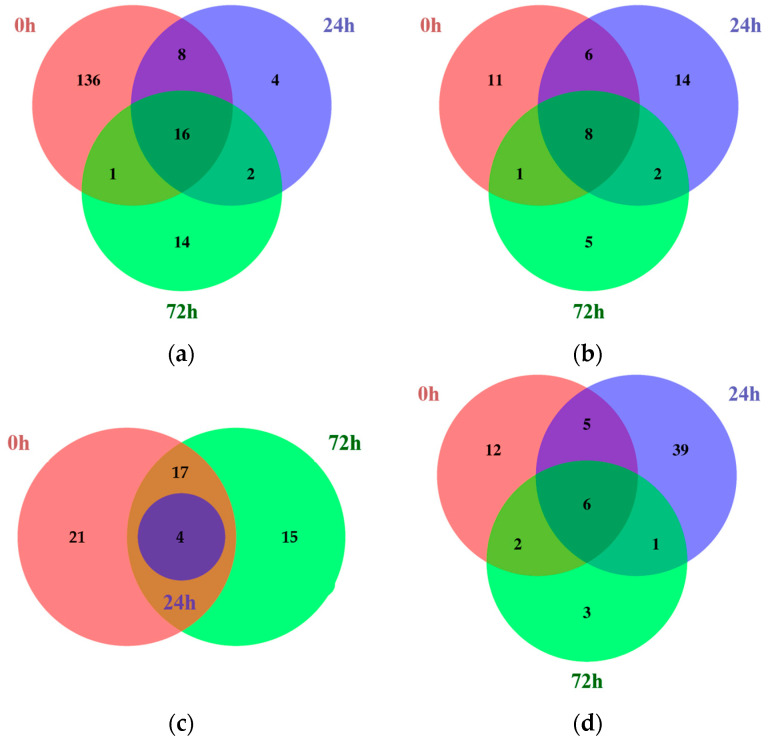
Venn diagrams showing the number of common DEGs identified comparing Kwangan and KA-1285 at each time point within each tissue under 200 mM NaCl. Upregulated (**a**) and downregulated (**b**) common DEGs in leaves. Upregulated (**c**) and downregulated (**d**) common DEGs in roots.

**Figure 2 plants-13-00254-f002:**
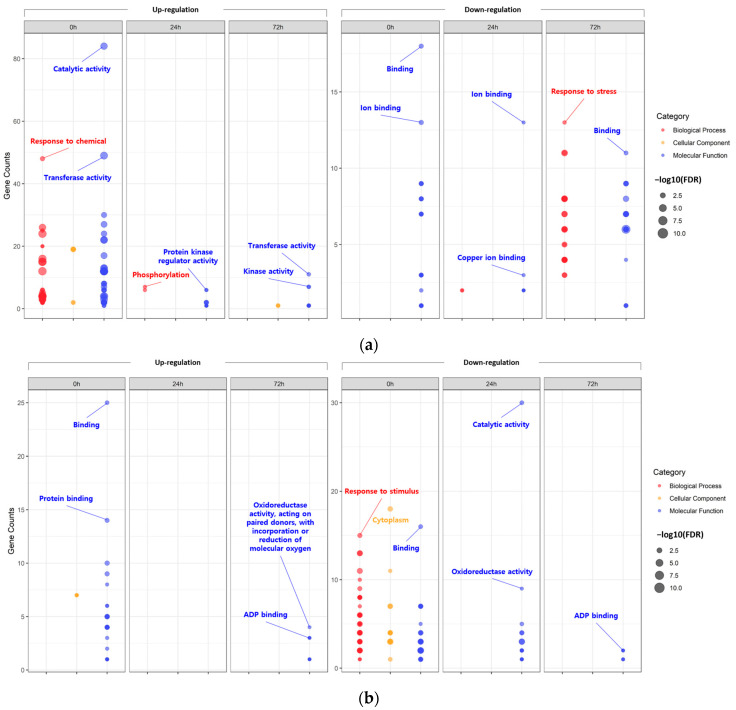
Bubble plot showing the enrichment results for gene ontology (GO). Enriched GO terms of differentially expressed genes (DEGs) identified in leaves (**a**) and roots (**b**).

**Figure 3 plants-13-00254-f003:**
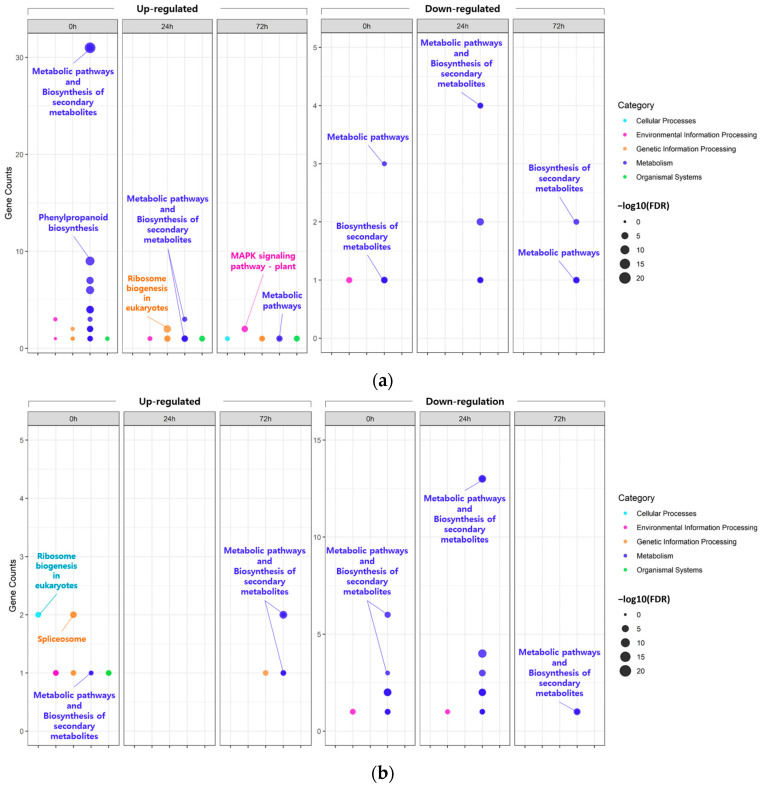
Bubble plot showing the enrichment results for Kyoto Encyclopedia Genes and Genomes (KEGG) pathway. KEGG pathway enrichment of DEGs identified in leaves (**a**) and roots (**b**).

**Figure 4 plants-13-00254-f004:**
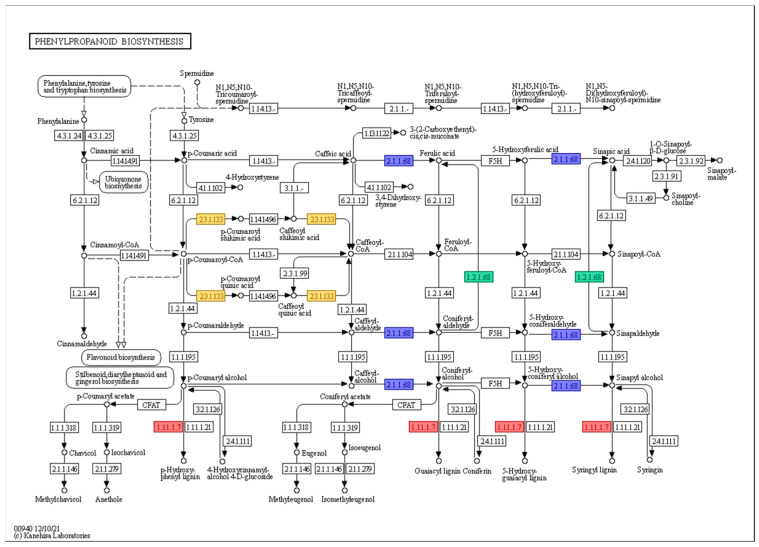
The phenylpropanoid biosynthesis pathway in response to 200 mM NaCl in soybean plants. The red, blue, green, and yellow boxes indicate that the genes were classified as ‘peroxidase (EC 1.11.1.7)’, ‘coniferyl-aldehyde dehydrogenase (EC 1.2.1.68)’, ‘caffeate O-methyltransferase (EC 2.1.1.68)’, and ‘shikimate O-hydroxycinnamoyltransferase (EC 2.3.1.133)’, respectively.

**Figure 5 plants-13-00254-f005:**
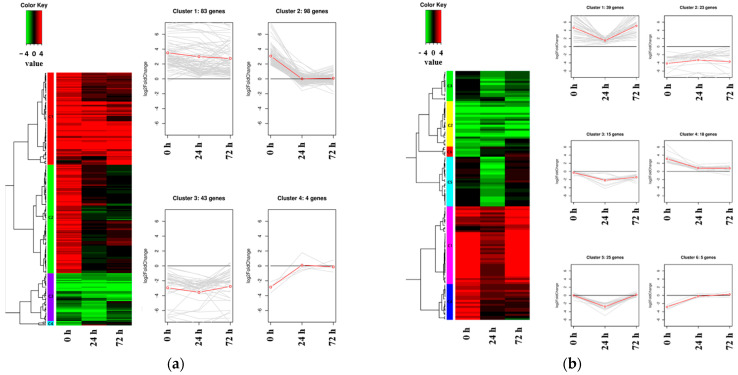
Heat map and line plot showing the expression levels and patterns of DEGs in each cluster. The red line shows the mean value of Log_2_ fold change (FC) in gene expression, as determined through RNA sequencing analysis. The cluster of DEGs identified in the leaves between the soybean origin cultivar ‘Kwangan’ and the salt-tolerant mutant induced by gamma-ray irradiation ‘KA-1285’ at three time points under 200 mM NaCl (**a**). The cluster of DEGs identified in the roots (**b**).

**Figure 6 plants-13-00254-f006:**
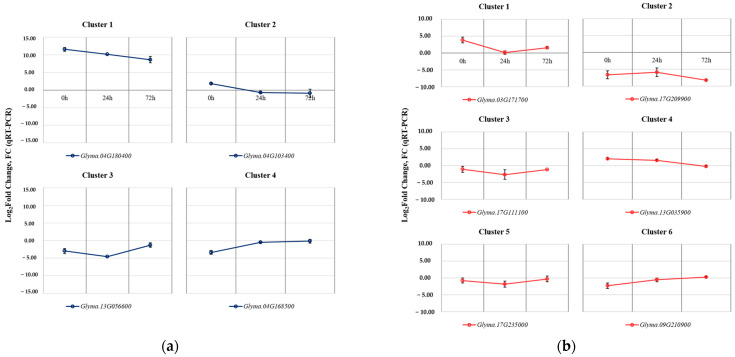
RT-qPCR results for the validation of selected DEGs with SNP mutations identified between Kwangan and KA-1285. The validation of selected DEGs in leaves (**a**). The validation of selected DEGs in roots (**b**). The error bars represent the standard deviations calculated from three biological replications.

**Figure 7 plants-13-00254-f007:**
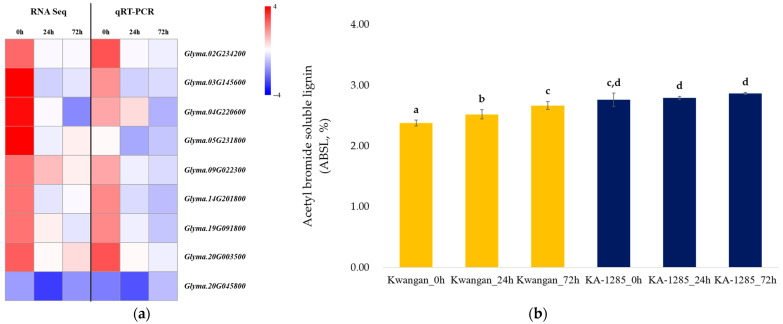
Validation of the DEGs related to the phenylpropanoid biosynthesis pathway. Heatmap showing the expression patterns of DEGs, comparing the results obtained from RNA sequencing analysis and RT-qPCR (**a**). Accumulation of acetyl bromide soluble lignin contents (ABSL) over time under 200 mM NaCl in Kwangan and KA-1285 (**b**). Different letters indicate statistically significant differences determined using the least significant difference (LSD test, *p* < 0.05). The error bars represent standard deviations calculated from five replications. The yellow bar represents the ASBL of Kwangan, while the dark blue represents the ASBL of KA-1285.

**Table 1 plants-13-00254-t001:** Number of differentially expressed genes (DEGs) identified through a comparison between the soybean cultivar ‘Kwangan’ and the salt-tolerant mutant ‘KA-1285’ at each time point within each tissue under 200 mM NaCl.

Tissues	Time Points	Number of DEGs		Total DEGs
		Upregulated	Downregulated	
Leaves	0 h	161	26	187
	24 h	30	30	60
	72 h	33	16	49
Roots	0 h	42	25	67
	24 h	4	51	55
	72 h	36	12	48

**Table 2 plants-13-00254-t002:** Functional annotations and expression patterns of the identified DEGs in the phenylpropanoid biosynthesis pathway compared between the Kwangan and KA-1285 under 200-mM-NaCl condition. The color in table indicates the level of gene expression from upregulation (red) to downregulation (green).

Gene ID	Log_2_ (Fold Change)						Annotation	EC * Number
	Leaf			Root				
	0 h	24 h	72 h	0 h	24 h	72 h		
*Glyma.02G234200.1*	2.51	−0.07	0.02	1.20	0.51	−0.05	Peroxidase superfamily protein	1.11.1.7
*Glyma.03G145600.1*	4.90	−0.69	−0.32	0.19	−0.60	−0.38	Peroxidase 2	1.11.1.7
*Glyma.04G220600.1*	3.78	0.07	−1.72	0.50	−0.31	−0.38	Peroxidase superfamily protein	1.11.1.7
*Glyma.05G231800.1*	4.38	−0.11	0.35	−0.25	−0.06	0.83	Aldehyde dehydrogenase 2C4	1.2.1.68
*Glyma.09G022300.1*	2.30	1.17	0.39	0.12	0.38	−0.21	Peroxidase 2	1.11.1.7
*Glyma.14G201800.1*	2.22	−0.30	−0.03	0.30	0.22	0.27	Peroxidase superfamily protein	1.11.1.7
*Glyma.19G091800.1*	2.36	0.25	−0.28	0.50	0.12	−0.28	Peroxidase superfamily protein	1.11.1.7
*Glyma.20G003500.1*	2.59	0.23	0.61	0.26	0.04	−0.01	O-methyltransferase 1	2.1.1.68
*Glyma.20G045800.1*	−1.54	−2.93	−1.62	0.63	−1.19	0.66	Spermidine hydroxycinnamoyl transferase	2.3.1.133

* EC, enzyme commission.

## Data Availability

The RNA-Seq datasets presented in this study were deposited on NCBI SRA database as NCBI BioProject numbers: PRJNA999740 (RNA-Seq of roots in Kwangankong), PRJNA999471 (RNA-Seq of leaves in Kwangankong), PRJNA999468 (RNA-Seq of roots in KA-1285), and PRJNA999461 (RNA-Seq of leaves in KA-1285).

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
