# Peer review of "Transcriptome Profiling of a Soybean Mutant with Salt Tolerance Induced by Gamma-ray Irradiation"

_plants, 2024, doi:10.3390/plants13020254_

Round 1
Reviewer 1 Report
Comments and Suggestions for Authors
In this study, Kwangan 'and' KA-1285 'were used as experimental materials to study the expression of differential genes in leaves and roots at 0h, 24h and 72h under 200 mM NaCl treatment by transcriptome analysis. Enrichment analysis showed that it was mainly distributed in Metabolic pathways and Biosynthesis of secondary metabolites. Compared with the control, the lignin content of KA-1285 leaves increased significantly under salt treatment, which may be a physiological reaction of soybean resistance to salt stress. I have the following questions about the results of this study:
1 In this study, why was transcriptome analysis performed at 0 h, 24 h, and 72 h after salt treatment?
2 Was the lignin content of roots measured in this study? Has anything changed?
3. The description in the abstract does not correspond to the conclusion of this paper, such as the results related to lignin.
4 The response of hormone pathways in plants under abiotic stress is relatively rapid. This aspect is less discussed in this study and should be supplemented.
Author Response
First of all, I would like to thank you for evaluating our manuscript on behalf of all of the authors. We've made some key rectifications based on the reviewer's feedback, please see the attachment.

Reviewer 2 Report
Comments and Suggestions for Authors
The present manuscript deals with the transcriptomic investigation of soybean cultivars under salinity stress. Although, the authors have conducted an extensive level of transcriptomic analysis, the manuscript lacks novelty and interpretation of results. Moreover, the authors need to add physiological and stress parameters upon salinity stress. Also, the transcriptome data needs more extensive analysis for example is there any mechanism hampered related to photosynthesis in leaf tissues whereas there are numerous transporters in roots which aids in plant recovery during salinity stress. Similarly, several ROS related genes apart form peroxidases also acts under abiotic stress, it will be interesting to know about their expressions.
In addition, there were reports suggesting the role of MYB TFs particularly in soybean upon salinity stress. Overall, the present manuscript has provided a comprehensive report on the transcriptomic analysis and lacks to eloborate scientific mechanism behind the DEGs to address tolerance to salinity.
I request the authors to utilize the transcriptome dataset to identify / investigate in-depth stress tolerance mechanisms or regulations/ candidates and correlate with the physiological parameters. This will benefit the readers and scientific community.
Comments on the Quality of English LanguageThe english language needs extensive revision and kindly modify "two locations" to "two tissues" throughout the manuscript.
Author Response

(The authors gave the same response as above.)

Reviewer 3 Report
Comments and Suggestions for Authors
The author conducted transcriptome sequencing on the salt tolerance of soybean germplasm(salt-tolerant soybean mutant induced by gamma-ray irradiation)and identified some important genes, laying the foundation for molecular breeding and breeding of new salt alkali tolerant soybean varieties.
Author Response

(The authors gave the same response as above.)

Reviewer 4 Report
Comments and Suggestions for Authors
This manuscript clearly showed the changes of mRNA after salt treatment in soybeans, and the research methods, research results, and paper writing were excellent. However, if necessary, the author needs to briefly describe in the abstract the relationship with the gene (GmSalt3) previously reported in KA-1285.
Author Response
- First of all, I would like to thank you for evaluating our manuscript on behalf of all of the authors. GmSalt3 is one of the major QTLs related to salt tolerance in soybeans and has been identified as a QTL related to ion accumulation in soybeans through various studies. This is briefly explained in Line 80 of the Introduction. However, when we reviewed the abstract, we found that these explanations were missing. Accordingly, we changed the abstract to state that we identified genes within the Major QTL region related to salt tolerance. please see the attachment.

Round 2
Reviewer 1 Report
Comments and Suggestions for Authors
The author only improved the language, and the data is still insufficient to support its publication in Plants.
Reviewer 2 Report
Comments and Suggestions for Authors
The authors have addressed and clarified most of the comments and queries. Further, the revised manuscript is in better form than the previous version. Based on the response and revisions I agree for the acceptance of this manuscript.